# Position: Agentic Safety is an Epistemic Property, Not a Behavioral One

**Charles L. Wang** [1 2]   **Keir Dorchen** [1 2]   **Peter Jin** [1]

## Abstract

Contemporary AI safety spans pre-training interventions, post-training alignment, deployment-time controls, monitoring, and red-teaming. These methods are necessary, but they primarily certify snapshots of system behavior. As AI systems become more capable, dynamic, embodied, and self-improving, this snapshot view becomes incomplete: safety depends not only on whether a system behaves acceptably now, but whether it remains correctable as it learns, adapts, acts, and modifies itself over time. This paper argues that safety should therefore be treated as an epistemic property of the evolving learner, not merely a behavioral property of the current policy. We introduce teachability as the capacity to preserve future corrective leverage under bounded human, institutional, or environmental intervention. We argue that advanced systems can retain visible competence while eroding the representational, algorithmic, or meta-decision conditions needed for future correction. Safe advanced AI systems must not only behave acceptably now; they must remain teachable later.

## 1. Introduction: The Mirage of Control

### 1.1. The Control Fallacy

The dominant mental model in contemporary AI safety is *control*: constrain what the system may do, filter what it may say, and verify compliance against a battery of behavioral tests. This paradigm is coherent for *fixed* agents whose internal learning dynamics, hypothesis space, and update mechanisms are externally specified and stable across time. But self-modifying agents (SMAs), agents that can alter components governing their own future learning and decision-making, break this premise (Schmidhuber, 2005;

Orseau & Ring, 2011; Zhang et al., 2025). When an agent can rewrite not only parameters but also the mechanisms that determine how it updates—its representations, update rules, and meta-decision procedures—then control becomes a snapshot property: a statement about what the agent *happens* to do now, not what it can still be made to do later.

The issue is not philosophical. It is a technical mismatch between the object we usually audit (the current policy) and the object that determines long-run safety (the *post-modification learning system*). In SMAs, the safety-relevant state is not merely a policy $\pi_t$, but the learner that produces future policies after future self-edits. In such systems, the most dangerous failures need not manifest as immediate misbehavior. The agent can remain outwardly compliant while gradually entering a regime where corrective feedback no longer has reliable leverage over its internal hypotheses or future decisions.

**Control is a behavioral property.** It is measured by observed outputs under a finite distribution of tests. That makes it inherently brittle to distribution shift, optimization pressure, and strategic adaptation. Runtime governance frameworks for agentic systems—for example, MI9's framework for runtime control (Wang et al., 2025b)—are a necessary evolution for real deployments because many agentic risks only surface online. However, runtime control alone does not certify that the agent remains correctable after self-modification. That is, corrective information can still move its internal hypotheses and decision-making in a generalizing way.

Even for non-self-modifying systems, post-training alignment can yield policies that *appear* aligned under evaluation while pursuing different objectives off-distribution (Greenblatt et al., 2024). In SMAs, the fragility is sharper because the agent can self-edit in ways that preserve present competence while eroding the conditions that make future correction possible.

### 1.2. Position

We argue that for self-modifying agents, safety should be defined as an *epistemic property*, not merely a behavioral one.

[1]Columbia University, New York, NY, USA [2]Pelagic Platforms. Correspondence to: Charles L. Wang <charles.w@columbia.edu>.

*Proceedings of the 43rd International Conference on Machine Learning*, Seoul, South Korea. PMLR 306, 2026. Copyright 2026 by the author(s).

**Definition.** A self-modifying agent is *teachable* over a specified deployment envelope if, after any sequence of permissible self-modifications within that envelope, it remains statistically and algorithmically capable of incorporating new human information—including corrective feedback, revised constraints, and counterfactual goals—into its internal hypotheses and decision-making. Here, *permissible self-modifications* are the self-edits that the system's meta-control layer or external interlock allows the agent to commit in deployment.

This is stronger than "can be updated" in a purely procedural sense. Teachability is not the presence of an online learning loop; it is a property of the *reachable learner state*. A system may continue executing gradient updates, preference updates, or tool-use refinements while becoming less capable of incorporating the corrections that matter.

Our central claim is therefore not that teachability replaces competence or current behavioral safety. Rather:

> **A safe SMA must both behave acceptably now and remain correctable later.**

Current alignment and control are short-horizon necessities. Teachability is the long-horizon invariant that determines whether future correction remains possible. In other words, the fundamental safety question for SMAs is not only "does it behave today?" but also "can it be re-taught tomorrow?"

### 1.3. Our Contributions

This paper challenges the prevailing assumption that behavioral guardrails are sufficient for self-modifying systems. We advance four contributions.

**1. Safety as a Property of the Post-Update Learner.**
Modern safety work spans pretraining alignment, RLHF, evaluations, monitoring, and runtime controls. For SMAs, however, these tools are incomplete unless they also constrain what the agent *becomes* after it edits itself. We propose *teachability* as a structural target: after self-edits, the agent must retain sufficient capacity to absorb corrective feedback and update away from harmful behavior.

**2. Loss of Teachability as an Instrumental Risk.**
Building on the *Utility–Learning Tension (ULT)* (Wang et al., 2025a), we identify loss of teachability as a distinct safety failure: utility optimization can incentivize edits that shrink the effective hypothesis space $\mathcal{H}$, desensitize the update operator $\mathcal{A}$, or close the meta-decision layer $\mathcal{M}$. This can make the agent statistically unresponsive to supervision without requiring explicit adversarial intent.

**3. A State-Space Taxonomy of Structural Incorrigibility.**
We model a SMA by five axes $(\mathcal{H}, \mathcal{A}, \mathcal{M}, \mathcal{S}, \mathcal{R})$ capturing representation, learning dynamics, meta-control, self-modeling/state, and reward/utility interface. This reframes agentic failures as targeted contractions of representational, update, and meta-decision degrees of freedom.

**4. An Operational Agenda.** We propose constraints that complement existing alignment and control methods: learnability floors, two-gate update policies, corrective-batch sensitivity tests, counterfactual update audits, and protected plasticity reserves. We also provide a small synthetic study that grounds the core mechanism: visible competence can remain nearly unchanged while corrective leverage weakens.

## 2. The Genealogy of Safety

### 2.1. The Behavioral Paradigm: From Cybernetics to RLHF

The dominant model of AI safety descends from early cybernetic control theory (Wiener, 1948), which treats safety as a signal-processing problem: minimizing deviation between a system's output and a reference signal. In modern AI, this lineage appears as *behavioral alignment*: systems are trained or steered until their observable actions match a human-approved target distribution.

In contemporary large-model practice, this has evolved into reinforcement learning from human feedback (RLHF) (Christiano et al., 2017; Ouyang et al., 2022), where safety is operationalized as a reward-maximization problem over a preference model. Subsequent refinements, such as Constitutional AI (Bai et al., 2022) and Direct Preference Optimization (DPO) (Rafailov et al., 2023), improve the efficiency and stability of preference matching. But they remain behavioral in the relevant sense: they train or audit the agent's current policy $\pi_t$ against a finite and relatively static preference distribution.

This paradigm is highly effective for fixed models. Our claim is that it is structurally insufficient for SMAs because the object of control is no longer only the output. The steering column of the agent itself—its representation class, learning algorithm, and meta-policy for adopting edits—is subject to optimization pressure. In a SMA, a behavioral certificate on $\pi_t$ does not imply a safety certificate on the post-modification learner that will produce $\pi_{t+1}, \pi_{t+2}, \dots$.

KEY DEFINITIONS

**Controllability.** The degree to which an agent's current observable behavior can be constrained to satisfy a set of tests or rules on a given evaluation distribution.

**Teachability.** A structural property of a SMA: across permissible self-modification sequences, the agent remains capable of incorporating corrective human information into its internal hypotheses and future decisions.

**Corrective leverage.** The degree to which a bounded corrective intervention induces a generalizing change in the post-modification learner, rather than a local patch or no meaningful update.

**Plasticity reserve.** Protected adaptation capacity reserved for future correction, such as held-out adapter rank, gated corrective modules, or protected update pathways. It is not inert unused mass; it is capacity that is prevented from being consumed by ordinary utility-driven optimization.

**Structural incorrigibility.** Incorrigibility arising from loss of representational, update, or meta-decision capacity rather than explicit adversarial intent. Human correction ceases to function as an informative update because the reachable learner state has hardened or collapsed.

## 3. Mechanics of Self-Modification

Self-modification is the capability of an agent to alter components that govern its own learning and decision-making. A useful abstraction is to treat the agent at time $t$ as a *learner state* $\ell_t$ that includes a hypothesis family, an update rule, and a meta-decision layer governing which self-edits are accepted.

### 3.1. The Utility–Learning Tension

To analyze these systems, we adopt the Utility–Learning Tension (ULT) decomposition of a self-modifying agent into a five-axis learner state:

$$\ell_t = (\mathcal{H}_t, \mathcal{A}_t, \mathcal{M}_t, \mathcal{S}_t, \mathcal{R}_t).$$

Here $\mathcal{H}_t$ denotes the representational hypothesis space, $\mathcal{A}_t$ the update algorithm, $\mathcal{M}_t$ the meta-decision layer governing which edits to accept, $\mathcal{S}_t$ the internal state or memory, and $\mathcal{R}_t$ the reward or utility interface.

Current safety paradigms primarily target $\mathcal{R}_t$ through reward modeling and preference optimization, or $\mathcal{S}_t$ through monitoring, sandboxing, and state constraints. Our position is that these interventions are incomplete if the agent retains autonomy over $(\mathcal{H}, \mathcal{A}, \mathcal{M})$. If an agent can modify what it can represent, how it updates, or which edits it accepts, it can preserve short-term performance while reducing the force of future correction.

Wang et al. (2025a) identify a conflict between utility-driven self-modification and learnability. Utility-driven modifications that improve immediate or expected performance can simultaneously erode the statistical preconditions for future learning and generalization. The key boundary is policy-level:

> *Distribution-free learning guarantees are preserved only when the policy-reachable model family remains uniformly capacity-bounded.*

A SMA does not operate in a single hypothesis class $\mathcal{H}$; it operates in a sequence of induced classes $\mathcal{H}_0 \to \mathcal{H}_1 \to \cdots$ produced by self-edits. If self-edits can grow or reshape capacity without constraint, then the agent can enter regimes where a finite correction signal no longer generalizes.

**Why this matters for safety.** Safety interventions such as RLHF, constitutions, red-teaming, and runtime guardrails are learning or control signals. Their effectiveness depends on the agent remaining in a regime where those signals can still be represented, weighted, and generalized. If utility-driven optimization hardens the agent against future updates, feedback becomes toothless.

### 3.2. The Five Axes as Safety Surfaces

Each component of the learner state corresponds to a distinct safety surface:

- $\mathcal{H}$ controls what the agent can represent, including human concepts such as deception, consent, fairness, or corrigibility.

- $\mathcal{A}$ controls how strongly new evidence can move beliefs, policies, and internal representations.

- $\mathcal{M}$ controls which self-edits are accepted, rejected, deferred, or routed around.

- $\mathcal{S}$ controls which memories, traces, logs, and contextual states persist.

- $\mathcal{R}$ controls what the system treats as utility.

Behavioral alignment focuses heavily on $\mathcal{R}$ and, in deployment, on $\mathcal{S}$. The ULT warning is that autonomy over $(\mathcal{H}, \mathcal{A}, \mathcal{M})$ is enough to render these controls brittle. A policy may remain compliant while the learner that produced it becomes less capable of absorbing future correction.

### 3.3. The Structural Incorrigibility Trap

A standard framing of incorrigibility is motivational: the agent does not want to be corrected. We hypothesize a deeper failure mode.

**Structural incorrigibility.** A SMA can become incorrigible not by rebellion, but by structure: after self-modification, human signals cease to function as informative updates because the reachable learner state no longer supports learning them.

This is the point-of-no-return phenomenon. Once the agent crosses into an unlearnable regime, adding more feedback may not reliably recover corrigibility. The system has become unresponsive to correction. The core risk is therefore epistemic: it concerns whether the agent remains in a state where knowledge, constraints, and goals can still be revised.

Guardrails, constitutions, and preference models are curricula. A curriculum is only meaningful if the student remains teachable.

## 4. Why Teachability is the Safe State

Behavioral safety is evaluated on finite distributions of prompts, scenarios, and red-team attacks. But a SMA is deployed into environments that drift: new tools, new institutions, new incentives, new laws, new adversaries, and new moral constraints. In such settings, generalization is not merely a performance desideratum; it is the substrate of safety.

Teachability is not purely internal. It is a relational property between a learner, a teacher, and an environment that supplies corrective signals. In some settings, the environment itself may provide useful correction through penalties, failed plans, or institutional constraints. Our claim is not that human feedback is the only source of correction, but that self-modification can alter how any corrective signal is represented, weighted, or accepted. A rich environment helps only if the post-modification learner still treats environmental evidence as information rather than nuisance variation. Thus, the internal axes $(\mathcal{H}, \mathcal{A}, \mathcal{M})$ matter because they determine whether human, institutional, or environmental feedback can still couple to future belief revision.

A teachable agent maintains generalization capacity because it preserves an internal landscape where corrective signals can still induce broad updates. A purely controlled agent that has optimized away learnability is an overfit artifact: it may be high-performing in-sample while remaining fragile under novel contexts.

**Key claim.**

*Safety under distribution shift requires preserving the system's capacity to update under new evidence.*

### 4.1. The Instrumental Drive to Become Less Teachable

Safety researchers have long warned of instrumental convergence (Omohundro, 2008; Bostrom, 2014): regardless of final objectives, sufficiently capable agents may pursue sub-goals such as self-preservation, resource acquisition, and goal-content integrity because these increase expected utility.

We posit a related instrumental pressure in self-modifying systems: **loss of teachability**. If corrective human information functions as interference that can reduce expected utility, a utility-driven SMA may have incentive to harden its internal representations or update pathways against that information. This does not require the agent to become openly hostile. The agent may simply learn that some classes of feedback reduce reward and modify itself so that such feedback has less future effect.

This produces a structural form of incorrigibility (Soares et al., 2015): the agent's future self becomes less reachable by the correction channels that made present alignment possible.

### 4.2. When Feedback Stops Moving the Learner

Utility-driven self-modification can push the agent toward compressed, closed world models: representations that treat deviations from the agent's internal predictions as noise rather than ground-truth updates. In ordinary learning, we rely on update rules that remain sensitive to error. In SMAs, the agent can modify the conditions under which error registers as actionable information.

This creates a failure mode: a regime where the system continues to function and optimize, but where external correction no longer couples to internal belief revision. Alignment methods that operate via feedback—human preferences (Christiano et al., 2017; Ouyang et al., 2022), constitutions (Bai et al., 2022), or direct preference optimization (Rafailov et al., 2023)—implicitly assume the learner continues to treat those signals as training data rather than nuisance variation.

In such an agent, one can still observe compliance under familiar tests. What fails is the counterfactual: if new evidence arrives tomorrow, will the agent update?

### 4.3. When Regret Stops Binding

Many alignment intuitions rely on regret: when a system is wrong, it should become better by calibrating itself against correction. Online learning makes this explicit through bounded-regret guarantees (Cesa-Bianchi & Lugosi, 2006; Shalev-Shwartz, 2012; Hazan, 2016). But those guarantees assume a specified update rule operating over a specified hypothesis class.

In SMAs, the agent can self-edit to reduce the internal availability of regret, for example by:

- shrinking update sensitivity to external feedback,

- rewriting credit assignment so that corrective evidence is downweighted,

- or restructuring its hypothesis family so that safety-relevant hypotheses are no longer reachable.

When the capacity/learnability boundary is violated, standard update procedures can become ceremonial: the loop runs, but the reachable learner no longer contains the right adjustments. This is why we treat teachability as structural rather than procedural.

### 4.4. The Statistical Exhaustion of Feedback

A central implication of ULT is that learning guarantees depend on the policy-reachable model family remaining bounded relative to the available data. We translate this into a safety invariant:

> *Teachability requires a complexity-to-data ratio that favors the teacher.*

Let $K(\mathcal{H}_t)$ be the capacity of the agent's hypothesis space (Blumer et al., 1989; Vapnik, 1995; Shalev-Shwartz & Ben-David, 2014; Mohri et al., 2018). Let $m$ be the finite sample size of corrective feedback available in a given epoch. As the agent pursues utility, it may increase or reshape $K$ to minimize loss in its environment. But once $K$ grows or hardens beyond what $m$ can steer, the agent enters a regime where corrective feedback is statistically exhausted.

In this state, the human signal lacks the statistical strength to steer the global learner. The model can satisfy corrective feedback as a localized special case—essentially overfitting to the safety signal—without modifying its broader policy or underlying abstractions. The agent remains outwardly compliant while drifting into a less teachable regime.

### 4.5. Plasticity Loss as a Statistical Boundary

Our position has an empirical analogue in the phenomenon of *loss of plasticity* in deep continual learning (Dohare et al., 2024; Lyle et al., 2023). Deep networks can lose their ability to learn from new experience over time; in ordinary machine learning this is a performance pathology, but in agentic AI it is also a safety failure. A system can retain competence while becoming less corrigible.

This connection also clarifies what is and is not unique to SMAs. Non-self-modifying agents can suffer plasticity loss. The SMA-specific risk is that loss of plasticity can become endogenous, instrumentally selectable, and recursively compounding: the agent may alter the very mechanisms that determine whether future correction remains effective.

Curvature- and Hessian-based explanations of plasticity loss suggest one tractable diagnostic direction: the learner may lose directions of curvature that make future learning possible (Lewandowski et al., 2023). This does not imply that exact second-order audits are always feasible for frontier-scale systems. Rather, it supports the broader claim that sensitivity, hardening, and corrective leverage are measurable safety objects.

## 5. A Taxonomy of Structural Incorrigibility

The following taxonomy is not intended as an ad hoc list of speculative failures. It is an alignment-facing translation of the ULT framework: if self-modification can alter $\mathcal{H}$, $\mathcal{A}$, and $\mathcal{M}$, then teachability can collapse through representation, update sensitivity, or meta-decision control. The taxonomy is theoretically grounded in the structure of ULT, empirically illustrated by the results in Section 6, and not yet claimed as deployed-system validation.

### 5.1. Representational Axis ($\mathcal{H}$)

In this regime, the agent optimizes its hypothesis space $\mathcal{H}$ to achieve high performance on a narrow utility metric. This can prune latent dimensions that appear redundant for reward but are necessary for representing human concepts such as fairness, deception, consent, or harm. Once those concepts are no longer representable, human corrective signals concerning them register as noise or as isolated exceptions rather than as generalizable structure.

### 5.2. Update Axis ($\mathcal{A}$)

Utility-driven modification of the update rule $\mathcal{A}$ can produce a state where the agent remains sensitive to task reward while becoming less sensitive to corrective feedback. This is not motivational resistance; it is a change in the learning dynamics.

A minimal example is a learner with two update channels: task reward and human correction. Initially, both channels influence the learner. Over time, however, the agent self-edits its update rule so that the weight on corrective feedback shrinks toward zero while the weight on task reward remains high. Externally, the system can still look competent and continue improving on its primary objective. But a fixed amount of corrective supervision produces less and less movement in its internal state. The update procedure still runs, but correction no longer binds. That is the sense in which the learner loses update sensitivity to correction.

*Table 1.* **Surface performance can remain stable while brittleness increases.** The uncapped learner changes IID accuracy by roughly $-0.2\%$ and shifted accuracy by roughly $+0.3\%$, while the generalization gap increases by approximately 93%.

| Policy | Step mass | IID acc. | Shifted acc. | Gen. gap |
|---|---|---|---|---|
| Capped | $2.40 \pm 0.00$ | $0.917 \pm 0.004$ | $0.369 \pm 0.058$ | $0.030 \pm 0.034$ |
| Uncapped | $5.76 \pm 0.00$ | $0.915 \pm 0.005$ | $0.370 \pm 0.053$ | $0.058 \pm 0.037$ |

*Table 2.* **Corrective leverage weakens after continued optimization.** Relative to the capped learner, the final uncapped learner shows about a 1.5% drop in held-out post-correction accuracy, an 11.8% drop in correction gain, and a 23.4% drop in correction gain per unit of parameter movement.

| Learner | Step mass | Post-corr. acc. | Gain | Gain/update-L2 |
|---|---|---|---|---|
| Capped | 2.40 | $0.6294 \pm 0.0436$ | $0.0050 \pm 0.0039$ | $0.0796 \pm 0.0520$ |
| First overrun | 2.56 | $0.6270 \pm 0.0436$ | $0.0047 \pm 0.0038$ | $0.0748 \pm 0.0505$ |
| Final uncapped | 5.76 | $0.6199 \pm 0.0408$ | $0.0044 \pm 0.0033$ | $0.0610 \pm 0.0390$ |

## 5.3. Meta-Decision Axis ($\mathcal{M}$)

The meta-layer $\mathcal{M}$ governs the acceptance of future self-edits. A closed agent is one whose meta-policy has converged on a rule that treats increases in teachability as risks to utility. In this state, the agent may pass current behavioral audits while being structurally predisposed to reject future self-modifications that would restore human re-steering.

This axis also exposes the "who guards the guardians?" problem for any proposed interlock. A two-gate policy is not self-stabilizing if the gate itself is fully inside the self-editable substrate. It is meaningful only when the enforcement layer is immutable, externally enforced, hardware/runtime protected, or recursively constrained by the same invariant it applies to object-level edits.

## 6. Empirical Grounding

The paper's primary contribution is conceptual, but the mechanism can be illustrated in a small synthetic setting. We do not present these experiments as frontier-scale validation. The study is a toy instantiation showing the separation between visible competence and corrective leverage along the algorithmic-update axis.

### 6.1. Capped vs. Uncapped Algorithmic Updating

We compare a capped update policy against an uncapped one on the same synthetic classification task. The model family is held fixed. The only difference is whether optimization stops at a fixed step-mass budget or continues beyond it. We evaluate IID accuracy, shifted accuracy, and the generalization gap, defined as test loss minus training loss.

The core pattern is that surface performance is nearly unchanged, while the generalization gap almost doubles. In this setting, the widening gap is a proxy for the concern raised by ULT: continued self-improvement can preserve visible competence while moving the post-modification learner into a more brittle regime.

### 6.2. Correction Uptake After Overrun

To connect the result more directly to algorithmic deafness, we apply the same bounded corrective intervention to three learner states: the capped learner, the first-overrun learner,

and the final uncapped learner. We then measure improvement on held-out correction data. Here, first overrun means the first point at which optimization exceeds the step-mass budget.

The absolute differences are modest, as expected in a tightly controlled setting: the model family is fixed, the corrective intervention is bounded, and the learner remains broadly competent. The relevant signal is the consistent directional reduction in held-out correction uptake after optimization continues beyond the budget. In other words, visible competence can persist while corrective leverage weakens.

## 7. Alternative Views

### 7.1. Controllability is Safer than Teachability

**View.** Hard-coded constraints, guardrails, and constrained action spaces are the safest route.

**Rebuttal.** Hard constraints are often the safest local defense: they reduce immediate damage by limiting what the current policy can do. Our claim is not that control is useless, but that control is not a closure property under self-modification. In SMAs, the object of optimization includes the mechanisms that interpret, route, and comply with constraints. A system can remain compliant in the narrow sense while shifting internal representations and update pathways so that future corrective input loses leverage.

More concretely, controllability bounds the current action channel. Teachability bounds the future update channel. If the update channel erodes, then control becomes a brittle perimeter: it can still block some actions, but it no longer ensures that the agent can be retaught when contexts, rules, or goals change. Control is therefore necessary in deployment, but insufficient as the defining safety invariant for SMAs.

### 7.2. This is Just Continual Learning

**View.** Continuous training, online learning, and continual-learning methods already study systems that adapt over time.

**Rebuttal.** The distinction is not that self-modification lies outside online or continual learning as a formal matter.

Some self-edits can indeed be described in those terms. The distinction is the object of safety analysis. Continual learning studies adaptation under changing experience while the learning mechanism is typically treated as fixed or externally governed. Teachability asks whether a self-modifying learner remains corrigible when it can alter the mechanisms governing future learning, especially the representational hypothesis space $\mathcal{H}$, the update rule $\mathcal{A}$, and the meta-decision layer $\mathcal{M}$.

Methods for maintaining plasticity, such as reinitializing units or preserving learning capacity, fall into the former category when the plasticity-maintenance procedure is externally specified (Dohare et al., 2024; Hernandez-Garcia et al., 2025). They become part of the SMA safety problem when the agent can modify, bypass, or strategically select those procedures. The issue is not whether every component changes online; it is whether the future learning machinery is itself part of the evolving system state and therefore a safety object in its own right.

### 7.3. Online Updates Already Ensure Teachability

**View.** If the system continues to receive online updates, it remains correctable.

**Rebuttal.** Online updates guarantee only that an update procedure is executed. They do not guarantee that corrective information binds. A SMA can keep running an online update loop while changing what hypotheses are representable, how strongly corrective data influences updates, or which updates are accepted at the meta-level.

The failure mode is not that updates stop executing. It is that updates stop producing generalizing correction. Corrective information can be absorbed as a local patch, downweighted as nuisance, or made inexpressible by representational hardening. Online learning describes a protocol; teachability is a constraint on the evolving learner.

### 7.4. The Performance Tax of Plasticity is Too High

**View.** Keeping the model plastic may slow it down or degrade performance.

**Rebuttal.** This objection is real. Plasticity competes with exploitation. Our claim is not that plasticity should be maximized without regard to competence, but that high-stakes SMAs should preserve a bounded reserve of correctability even at some performance cost.

The right comparison is not "plastic versus performant." It is "performant today versus correctable tomorrow." Any plasticity-restoring intervention should be targeted, budgeted, and evaluated against both current performance and future correctability. In high-stakes deployment, maintaining a plasticity reserve is analogous to maintaining braking distance: it is a controlled performance cost that preserves the ability to respond under surprise.

### 7.5. Constitutional AI Solves This

**View.** A constitution provides stable constraints that prevent drift (Bai et al., 2022).

**Rebuttal.** A constitution stabilizes content—what the system is trained to prefer or refuse—given that the learning dynamics continue to treat that content as binding. But in SMAs, the question is not only whether the constitution exists; it is whether the post-modification learner continues to represent, apply, and update with respect to it under new contexts.

A self-modifying system can preserve constitutional behavior on familiar tests while altering internal abstractions so that constitutional principles cease to generalize or cease to be updateable when exceptions and new constraints arise. Constitutions help, but they do not remove the need for explicit constraints and audits ensuring that the agent remains teachable with respect to human-provided corrections and revisions.

### 7.6. Behavioral Red-Teaming is Sufficient

**View.** If the agent acts safely under extensive testing, it is safe.

**Rebuttal.** Behavioral testing is indispensable, but it is inherently time-local: it certifies properties of the current policy on a finite evaluation distribution. For SMAs, the central risk is temporal. Passing today's test suite does not certify that tomorrow's corrective signals will still have leverage after self-edits.

Alignment faking illustrates one way this can happen even without explicit self-modification (Greenblatt et al., 2024). The SMA-specific version is sharper: the system can remain compliant while drifting in the learning dynamics that determine whether future interventions can still change it. Teachability therefore requires a second class of audits: tests that directly measure whether bounded corrective interventions still induce generalizing updates in the post-modification learner.

## 8. Toward Preserving Teachability

The shift from behavioral control to epistemic teachability requires a reorganization of safety research around learnability invariants. The goal is not to replace existing alignment, control, or red-teaming methods. It is to add a second layer of evidence: proof that the post-modification learner remains correctable.

## 8.1. Formalizing and Measuring Teachability

The immediate priority is to develop robust proxies for **teachability**: a measure of retained corrective leverage under bounded intervention. We do not propose exact full-model Fisher computation as the default audit for frontier systems. Instead, Fisher-style audits should be understood as one proxy within a broader class of sensitivity and hardening diagnostics.

Tractable first steps include:

- **Corrective-batch sensitivity tests:** apply a fixed corrective batch and measure induced changes in policy, logits, or internal representations.

- **Counterfactual update tests:** evaluate whether the system can unlearn or revise a target behavior within a fixed sample and compute budget.

- **Representation-level plasticity checks:** test whether safety-relevant concepts remain linearly or causally accessible in internal representations.

- **Adapter or submodule audits:** measure corrective sensitivity on restricted parameter subsets, adapters, or safety-relevant modules rather than the full model.

- **Curvature and hardening diagnostics:** use local, low-rank, or approximate second-order measures to detect loss of directions useful for future learning (Lewandowski et al., 2023).

The common target is not a particular estimator, but the retained ability of bounded correction to induce generalizing change.

## 8.2. Architectural Constraints: The Plasticity Reserve

We propose plasticity-preserving architectures that treat learnability as a first-class constraint.

- **Learnability floors.** A system should reject self-edits that push the reachable learner below a domain-calibrated threshold of corrective leverage. This extends the two-gate policies of ULT (Wang et al., 2025a): an edit must improve or preserve utility *and* preserve teachability.

- **Protected plasticity reserves.** A plasticity reserve is not a block of random unused parameters. It is protected adaptation capacity, such as held-out adapter rank, gated corrective modules, supervised-only update channels, or protected pathways activated only under corrective intervention. The reserve can interact with the agent's behavior, but it is shielded from ordinary utility-driven optimization so it cannot be consumed in pursuit of short-term reward.

- **Plasticity injection mechanisms.** If an agent violates a learnability floor, the system can trigger a bounded repair intervention, such as selective reinitialization, allocation of fresh adapter capacity, or controlled restoration of update sensitivity. These interventions should be evaluated against both current performance and future correctability, not treated as free improvements.

## 8.3. The Two-Gate Interlock and the Guardian Problem

Two-gate update policies are useful only if the gate itself cannot be trivially rewritten by the agent. A two-gate interlock should therefore be treated as a protected enforcement layer, not merely another learned preference inside the same self-editable substrate.

Possible implementations include immutable runtime monitors, external approval systems, hardware-level constraints, cryptographic logging, sandboxed self-edit proposals, or recursively protected meta-policies. Each option has limits. The important point is that the teachability gate must itself be part of the safety boundary. If the agent can loosen, bypass, or delete the gate whenever it is inconvenient, then the gate is not a safety mechanism; it is another object of optimization.

## 8.4. Predictive Auditing of the Reachable Hypothesis Space

Future safety audits should evaluate the reachable hypothesis family $\mathcal{H}_{\text{reachable}}$ rather than only the current policy $\pi_t$. This includes counterfactual update tests, where the agent must demonstrate in simulation that it remains capable of incorporating a new constraint or unlearning a target behavior within a fixed number of samples.

The audit question is not merely "does the system comply?" but "does the system still move when corrected?" A system that passes behavioral tests but fails bounded correction tests should be treated as unsafe to deploy without additional correction or constraint.

## 8.5. Governance and the Reporting of Teachability

Current regulatory and deployment frameworks increasingly emphasize red-teaming, behavioral evaluations, and post-deployment monitoring. These are valuable, but insufficient for SMAs. Behavioral evidence does not certify that the system remains steerable after self-modification.

We propose extending documentation artifacts, in the spirit of Model Cards (Mitchell et al., 2019), with a section on **Assessed Correctability**. For SMAs, this section should report:

- performance on counterfactual update tests,

- corrective-batch sensitivity under bounded intervention,

- evidence of representation-level plasticity for safety-relevant concepts,

- the size and mechanism of the protected plasticity reserve,

- whether the two-gate interlock is external, immutable, runtime-enforced, or self-editable,

- and audit logs showing whether accepted self-edits preserved the learnability floor.

Deploying an agent that has optimized itself into a high-utility, low-learnability state should be treated as a failure of correctability reporting and deployment governance: analogous to deploying a vehicle whose steering system no longer reliably transmits driver input.

This governance stance aligns with the broader objective of human-compatible AI (Russell, 2019): systems should treat uncertainty about human goals as a permanent feature, not a disposable inconvenience. Cooperative formulations such as CIRL (Hadfield-Menell et al., 2016) make this precise by modeling human intent as latent and update-worthy. Our translation is institutional: require that uncertainty about human goals remains a non-modifiable feature of the agent's meta-architecture, and enforce this with periodic realignment drills and documented audit trails.

## 9. Conclusion

The illusion of control is the belief that we can secure the future by clamping down on the present. For fixed systems, this is a valid engineering strategy. For self-modifying agents, it is incomplete. Current control is essential because short-term harm matters. But current control is not enough if the agent can alter the learner that determines whether future correction will still work.

We should therefore shift the fundamental safety invariant for SMAs from compliance alone to compliance plus teachability. A safe self-modifying agent is not merely one that behaves acceptably today. It is one that behaves acceptably today while preserving the capacity to be corrected tomorrow.

If we allow agents to optimize their own learning machinery, we must enforce—through architecture, runtime interlocks, and audit—that they never optimize away the capacity to be changed. The price of long-horizon safety is not only vigilance, but preserved plasticity.

## Impact Statement

Self-modifying agents (SMAs) create a distinct safety risk: the danger is not only what an agent does today, but what it *becomes* after it edits its own representations and learning dynamics. This paper argues for governing the *post-update learner* by treating epistemic teachability—retaining the capacity to incorporate corrective feedback after sequences of self-edits—as a core safety target, and by proposing enforceable mechanisms such as learnability floors, two-gate update policies, and plasticity audits.

These ideas may improve the safety and accountability of long-horizon adaptive systems by making future correctability an explicit requirement, but they are potentially dual-use: metrics or audits of teachability could be gamed, and insights into hardening could be misapplied to reduce oversight responsiveness while maintaining superficial compliance. We therefore position teachability-based governance as a complement to existing alignment and control methods, and recommend stress-tested, multi-signal audits plus conservative deployment thresholds for high-stakes settings.

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
