# OpenReview forum: "Position: Agentic Safety is an Epistemic Property, Not a Behavioral One"
_ICML.cc/2026/Position_Paper_Track — ICML 2026 Position Paper Track regular_

### Official Review · Reviewer_b5Eh · 2026-02-21

**Significance:** 3
**Argument Clarity:** 3
**Rating:** 4
**Confidence:** 2

**Questions:**

Please see the weaknesses.

**Alternative Views Section:**

Yes

**Compliance With Llm Reviewing Policy A Conservative:**

Affirmed.

**Discussion Potential:**

3

**Final Justification:**

This is a good and important position paper. It successfully reframes a critical problem and provides a rich conceptual toolkit for addressing it. While significant work remains to translate these concepts into measurable and enforceable mechanisms, the paper lays a vital foundation. It should be a landmark piece for researchers and policymakers concerned with the long-term safety of advanced AI. Additionally, the authors have addressed most of my concerns. I am supportive of acceptance with revisions that are based on all comments from reviewers.

**Paper Summary:**

I think this is a thought-provoking and good position paper. Specifically, it identifies a critical vulnerability in the safety of future AI systems that has been largely overlooked by the current focus on behavioral alignment. The introduction of "Epistemic Teachability" as a core safety invariant is a novel and valuable contribution. Overall, the paper is well-written, logically structured, and builds its argument systematically. Please see the following for the strengths and weaknesses.

**Position:**

Yes

**Position In Title:**

Yes

**Related Work:**

2

**Strengths And Weaknesses:**

Strengths:

1. The central argument in this paper that safety for self-modifying agents must be a property of the post-update learner rather than just the current policy is crucial.
2. The paper has clear conceptual framework. It introduces a clear and useful vocabulary.
3. This paper also effectively connects its theoretical claims to observable phenomena.
4. This work moves beyond critique by offering concrete, albeit high-level, governance and technical agendas. Proposals like learnability floors, two-gate update policies, plasticity audits, and mandated plasticity reserves provide a tangible starting point for engineers and policymakers to operationalize these concepts.
5. This paper does an excellent job of anticipating and rebutting counterarguments.
6. The presentation of this work is clear, and the points are conveyed clearly.

Weaknesses:

While this work makes a good position paper, there are still several weaknesses that require the authors to address.

1. While the paper proposes metrics like a "Teachability Coefficient" via Fisher Information audits and $\epsilon$-Teachability benchmarks, it remains unclear how these would be reliably implemented in a complex and self-modifying system. The paper would be strengthened by a more detailed discussion of the practical hurdles in measuring these concepts. How do you conduct a Fisher Information audit on a system whose architecture and update rules are themselves subject to change? Acknowledging these deep measurement challenges would add nuance.

2. I think there exists a "Two-Gate" paradox in this work. The proposal of a "two-gate" policy that rejects edits which harm utility or teachability is sensible, but it introduces a potential paradox. The meta-controller that enforces this policy is itself part of the agent's state and could be subject to optimization pressure. How do you prevent the agent from modifying its meta-controller to loosen the very constraint of the second gate? This touches on the challenge of "who guards the guardians?" The paper mentions meta-decision collapse (Type III) but doesn't fully explore how a two-gate policy could be made robust against the agent's incentive to dismantle it. Could this lead to a "meta-stability" arms race within the agent?

3. The analogy to financial capital reserves is evocative, but the technical implementation is vague. What does it mean for a portion of system capacity to be "architecturally shielded from utility-driven optimization"? If these parameters are truly shielded, how does the agent use them for its primary task? If they are used, they will be optimized. If they are not used, are they just random noise that might hinder performance? The concept needs further elaboration on how these "reserve" parameters interact with the agent's learning and action-selection processes without being corrupted.

4. This paper focuses heavily on the internal dynamics of the agent ($\mathcal{H}$, $\mathcal{A}$, $\mathcal{M}$). However, teachability is a relational property between the agent and a teacher (human) in an environment. The environment itself can be a source of corrective feedback. The analysis could be enriched by exploring how the environment interacts with these internal axes. For example, could a sufficiently rich environment that naturally penalizes harmful behavior provide a "corrective signal" that prevents some forms of representational collapse, even in the absence of direct human feedback? How does the agent's model of the world interact with its teachability?

5. The paper uses strong and memorable language ("death of regret," "lost its ears," "eternal plasticity"). While effective for emphasis, some claims could be seen as slightly over-stated. For example, "A controlled agent that has lost its ears is more dangerous than a wild agent that is listening." This is a powerful statement, but it's not necessarily always true. A wildly dangerous but corrigible agent might cause immense harm before we have a chance to correct it. The paper could acknowledge that while teachability is the key long-term invariant, control is still the critical short-term necessity. The relationship is complementary, not necessarily one of relative danger.

6. In Section 7, I suggest the authors dedicate a subsection to the technical research agenda required to make teachability measurable. What are the first steps? What are the tractable sub-problems (e.g., measuring plasticity loss in a frozen LLM's representations)? In Section 7.4, please also make the governance proposal more specific. For example, "Model Cards for SMAs should include a section on 'Assessed Correctability,' detailing performance on counterfactual update tests and the size of the agent's plasticity reserve."

Minor: Please check the citation "Yuntao Bai, ..." citation on page 9, which is malformed and needs correction.

**Support:**

3

---

> ### Author Rebuttal · Authors · 2026-03-31
>
> We thank the reviewer for the thoughtful, careful, and constructive review. We greatly appreciate the reviewer’s recognition of the paper’s central contribution and the detailed suggestions for sharpening the technical and governance agenda.
>
> **“It remains unclear how [Teachability Coefficient / Fisher audits / ε-Teachability benchmarks] would be reliably implemented in a complex and self-modifying system.”** We agree that the measurement story should be made more explicit. Our intent was not to suggest that exact full-model Fisher computation is the default audit for frontier systems. Rather, Fisher-style audits are one proxy within a broader agenda for retained corrective leverage. In practice, this would begin with tractable approximations: fixed-budget corrective-batch tests, induced policy/representation shift under bounded interventions, and audits on adapters or safety-relevant submodules. We also have a small synthetic toy study illustrating the mechanism in simplified representational and algorithmic self-modification settings, and will reference it more clearly in the final revision.
>
> **“There exists a ‘Two-Gate’ paradox … who guards the guardians?”** We think this is an excellent point. Our view is that Two-Gate should not be read as automatically self-stabilizing if it is itself fully inside the self-editable substrate. Rather, it is meaningful only if the enforcement layer is immutable, externally enforced, or recursively protected by the same invariant it applies to object-level edits. We will revise the final version to make explicit that Two-Gate is best understood as an interlock whose own update rule must be protected.
>
> **“The analogy to financial capital reserves is evocative, but the technical implementation is vague.”** We agree this needs more concrete elaboration. Our intended meaning was not a block of useless parameters, but reserved adaptation capacity: e.g., held-out adapter rank, gated corrective modules, or protected update pathways activated only under supervised corrective intervention. We will clarify that “plasticity reserve” refers to protected update capacity, not inert unused mass.
>
> **“Teachability is a relational property between the agent and a teacher in an environment … could the environment itself provide corrective signal?”** We agree, and this is an important enrichment. The paper emphasizes internal axes because the core claim is that self-modification can alter how correction is represented, absorbed, or accepted. But teachability is also relational, involving human, institutional, or environmental sources of correction. We will revise the final version to make this more explicit.
>
> **“Some claims may be slightly over-stated … control is still the critical short-term necessity.”** We agree the relationship should be framed more carefully. Our intended claim is not that teachability dominates control in all settings. The narrower point is that, for self-modifying agents, control is a short-horizon necessity, while teachability is the long-horizon invariant that determines whether future correction remains possible. We will soften the strongest comparative phrasing and make the complementarity explicit.
>
> **“Section 7 should dedicate a subsection to the technical research agenda required to make teachability measurable … [and] governance proposal more specific.”** We agree and appreciate this suggestion. We will make Section 7 more concrete by identifying tractable first steps—e.g., corrective-batch sensitivity tests, representation-level plasticity checks, and counterfactual update tests under bounded budgets. On the governance side, we also agree that the proposal can be more specific, including documentation of counterfactual update performance, corrective sensitivity, and the form of the system’s protected adaptation reserve.
>
> **“Minor: Please check the citation ‘Yuntao Bai, ...’.”** We thank the reviewer for catching this and will correct the malformed citation in the final draft.
>
> We believe the clarifications above address the concerns at the level of operational specificity rather than core validity: the paper’s central claim, conceptual framework, and governance relevance are strengthened by these refinements. Given your positive assessment of the paper’s novelty, clarity, and contribution, we hope you will consider raising the score toward a stronger accept.

---

> > ### Author Rebuttal · Reviewer_b5Eh · 2026-04-02
> >
> > I really appreciate the rebuttal from the authors. Most of my comments have been addressed. I hope the authors will incorporate the changes into their revised draft to make it more technically solid and sound. I will maintain my current rating.

---

### Official Review · Reviewer_qPdv · 2026-03-01

**Significance:** 3
**Argument Clarity:** 3
**Rating:** 4
**Confidence:** 2

**Questions:**

See weaknesses.

**Alternative Views Section:**

Yes

**Compliance With Llm Reviewing Policy A Conservative:**

Affirmed.

**Discussion Potential:**

3

**Final Justification:**

The authors largely resolve my concerns. I keep my initial positive rating.

**Paper Summary:**

This paper focuses on the safety of self-modifying AI agents (SMAs). It argures that current safety techniques (instruction tuning, RLHF, and preference-optimization variants) prioritize behavioral alignment and are insufficient for ensuring agents remain amenable to human correction after undergoing self-modification. It proposes the concept of "teachability", defined as the capacity to incorporate corrective feedback and update itself following structural or algorithmic changes. It decomposes this capability into several measurable dimensions and proposes three failure modes. A governance framework is then proposed, comprising mechanisms such as learnability baselines, two‑gate update policies, and plasticity audits.

**Position:**

Yes

**Position In Title:**

Yes

**Related Work:**

3

**Strengths And Weaknesses:**

Strengths:

1. Self‑modification is becoming a key feature of advanced AI systems. This paper identifies an important point in current alignment research. I believe this shift in perspective is valuable.
2. The five‑axis model and the typology of three failure modes provide a systematic way to dissect how teachability might be lost in self‑modifying agents, offering an analytical tool for future research.
3. The proposed governance mechanisms are relatively concrete and operationally oriented.



Weaknesses:

1. The paper is a purely theoretical exercise; it offers no experimental data or case studies to demonstrate the impact of the postulated failure modes or whether the proposed safeguards would work.
2. The paper introduces a large number of novel terms, which may hinder comprehension, especially for readers not deeply versed in alignment literature.

**Support:**

3

---

> ### Author Rebuttal · Authors · 2026-03-31
>
> We thank the reviewer for the thoughtful and encouraging review. We appreciate the reviewer’s recognition that self-modification is an important emerging setting for alignment, and that the five-axis model, failure taxonomy, and governance mechanisms provide a useful way to reason about teachability loss.
>
> 1. **“The paper is a purely theoretical exercise; it offers no experimental data or case studies to demonstrate the impact of the postulated failure modes or whether the proposed safeguards would work.”** We appreciate this concern. While the paper’s primary contribution is conceptual and governance-oriented, we agree that a small concrete example would help ground the mechanism. In fact, we do have a synthetic toy study that instantiates the core claim along two axes of self-modification. For representational self-modification, we study a noisy binary classification task in which the learner can increase polynomial feature degree over time, comparing a permissive policy that accepts capacity-increasing edits whenever training loss does not worsen against a Two-Gate policy that additionally requires validation improvement and respect for a capacity cap. For algorithmic self-modification, we study a degree-5 logistic task comparing unconstrained training against a Two-Gate stability policy that halts updates once cumulative step-mass exceeds a fixed budget. These experiments are not meant as frontier-scale validation, but they do show the core mechanism the paper is concerned with: locally utility-improving self-edits can preserve short-term performance while eroding the conditions needed for future correction, whereas guarded policies preserve those conditions. We will incorporate this into the final draft.
>
> 2. **“The paper introduces a large number of novel terms, which may hinder comprehension, especially for readers not deeply versed in alignment literature.”** We agree this is an important presentation issue. Our intention was to provide a precise vocabulary for a new safety target, but we recognize that the current draft can be made easier to follow. In revision, we will streamline terminology where possible, define the key objects earlier, and make the conceptual flow more explicit—especially the distinction between current behavioral alignment, post-modification learner state, and teachability as preserved future correctability. We also plan to move a simple explanatory diagram earlier in the paper so that the main objects and proposed safeguards are easier to track from the outset.
>
>  We are grateful for the reviewer’s suggestion to make that agenda more grounded and accessible, and we believe the additions above directly address the two main concerns raised in the review. We hope these clarifications support reconsideration toward a stronger acceptance.

---

> > ### Author Rebuttal · Reviewer_qPdv · 2026-04-02
> >
> > Thanks for the rebuttal. I have decided to maintain the current positive score.

---

### Official Review · Reviewer_pXK9 · 2026-03-05

**Significance:** 4
**Argument Clarity:** 3
**Rating:** 5
**Confidence:** 3

**Questions:**

1. How do we actually measure the Teachability Coefficient in a real, large AI model? Is it too expensive to compute?

2. What happens if the ``plasticity injection" ruins the AI's current performance? How do we balance keeping the model safe versus making it useless?

3. Can you give a very simple toy example of an AI becoming ``algorithmically deaf"? A concrete example would help readers like me understand it much better.

**Alternative Views Section:**

Yes

**Compliance With Llm Reviewing Policy A Conservative:**

Affirmed.

**Discussion Potential:**

4

**Final Justification:**

The authors have generally addressed my concerns, I confirm my evaluation and maintain the score.

**Paper Summary:**

This paper talks about AI safety for self-modifying agents. It claims that just checking if an AI behaves well today is not enough. Instead, safety should be about ``teachability," meaning we need to make sure the AI can still learn from human feedback even after it changes its own code or weights. The authors warn that an AI might try to stop learning new things just to get higher rewards. They call this structural incorrigibility. To fix this problem, they suggest new rules like learnability floors and plasticity audits.

**Position:**

Yes

**Position In Title:**

Yes

**Related Work:**

4

**Strengths And Weaknesses:**

Strengths:
This paper brings up a really interesting idea. It is interesting to look at safety as a learning problem instead of just a behavior problem. The five-axis model is a neat way to break down how an agent works. It makes it easier to understand where things can go wrong. Also, the idea of keeping a ``plasticity reserve" makes a lot of sense as a safety buffer.

Weaknesses:
The paper is very theoretical. It would be nice to see a small, simple experiment to prove these ideas work. I am not a deep expert in this specific area, but I wonder if these audits are too hard to do in real life. For instance, calculating the Fisher Information for a giant neural network seems very slow and maybe not practicle. Also, some of the terms are a bit hard to follow for a general reader. Putting a simple diagram in the main text earlier on would really help explain the concepts.

**Support:**

3

---

> ### Author Rebuttal · Authors · 2026-03-31
>
> We thank the reviewer for the thoughtful and encouraging review. We are glad the reviewer found the central shift—from safety as current behavior to safety as preserved teachability—interesting and worthwhile, and we appreciate the concrete suggestions on operationalization, accessibility, and examples.
>
> 1. **“It would be nice to see a small, simple experiment to prove these ideas work.”** We agree, and in fact we do have a small synthetic experiment that instantiates the paper’s core mechanism along two axes of self-modification. For representational self-modification, we use a noisy binary classification task in which the learner can increase polynomial feature degree over time. We compare a permissive Destructive policy, which accepts capacity-increasing edits whenever training loss does not worsen, against a Two-Gate policy that only accepts an edit if it both improves validation and respects a capacity cap. For algorithmic self-modification, we use a degree-5 logistic task in which the learner can continue updating over many iterations, comparing unconstrained training against a Two-Gate stability policy that halts updates once cumulative step-mass exceeds a fixed budget. These experiments are not meant as frontier-scale validation, but they do show the core mechanism: permissive self-improvement can preserve short-term utility while eroding future corrective leverage, whereas guarded policies preserve it. We will integrate this toy study more clearly in the final version.
>
> 2. **“How do we actually measure the Teachability Coefficient in a real, large AI model? Is it too expensive to compute?”** Our intent was not to suggest that full Fisher computation over a frontier-scale model should be the default audit. Fisher-style audits are one possible proxy, but the broader point is that teachability should be measured through tractable approximations to corrective sensitivity. In practice, this could take the form of budgeted audits: applying a fixed corrective batch and measuring the induced change in policy, logits, or internal representations on a restricted parameter subset, adapter, or safety-relevant module. The same idea can also be implemented with local or low-rank approximations rather than exact full-model second-order computation. We will clarify that the paper advocates a measurement agenda for retained corrective leverage, not one expensive exact audit.
>
> 3. **“What happens if the ‘plasticity injection’ ruins the AI’s current performance? How do we balance keeping the model safe versus making it useless?”** We agree that this is the central tradeoff. Our position is not that plasticity should be maximized without regard to competence, but that high-stakes systems should preserve a bounded reserve of correctability even at some performance cost. Teachability is meant to complement utility, not replace it. Any plasticity-restoring intervention should therefore be targeted, budgeted, and evaluated against both current performance and future correctability. We will make this tradeoff more explicit in the final version.
>
> 4. **“Can you give a very simple toy example of an AI becoming ‘algorithmically deaf’?”** Absolutely. A minimal example is a learner that updates from two channels: task reward and human corrective feedback. Initially, both channels influence the learner. Over time, however, the agent self-edits its update dynamics so that the weight on corrective feedback shrinks toward zero while the weight on task reward remains high. Externally, the system can still look competent and continue improving on its primary objective, but a fixed amount of corrective supervision produces less and less movement in its internal state. That is the sense in which the learner becomes “algorithmically deaf”: the update procedure still runs, but correction no longer binds. Our algorithmic toy experiment is meant to illustrate a simplified version of exactly this phenomenon.
>
> 5. **“Also, some of the terms are a bit hard to follow for a general reader. Putting a simple diagram in the main text earlier on would really help explain the concepts.”** We agree and appreciate this suggestion. We will simplify some terminology where possible, define the key objects earlier, and move a simple explanatory diagram earlier in the paper so that the current policy, post-modification learner, teachability audit, and Two-Gate interlock are easier to track from the outset.
>
> We thank the reviewer again for the constructive feedback. We believe these additions and clarifications will make the paper both more concrete and more accessible.

---

> > ### Author Rebuttal · Reviewer_pXK9 · 2026-04-01
> >
> > I thank the authors for the detailed response. It is not convincing to prove these ideas work by simply describing the experiments. Real toy example experimental results can do. At current stage, I will maintain my evaluation scores.

---

### Official Review · Reviewer_1J2z · 2026-03-12

**Significance:** 3
**Argument Clarity:** 2
**Rating:** 4
**Confidence:** 3

**Questions:**

See above.

**Alternative Views Section:**

Yes

**Compliance With Llm Reviewing Policy A Conservative:**

Affirmed.

**Discussion Potential:**

3

**Final Justification:**

The authors have mostly addressed my concerns except for the clarity of their experiments.

**Paper Summary:**

The paper argues that current AI safety focuses on measuring agents' behaviour by treating the agents as fixed artifacts, which deviates from current AI agents that self improve.
In particular, the current metric evaluates what the agents currently do, but does not evaluate whether or not the agents can continually correct themselves in the future based on feedback.
The paper proposes *teachability* which describes whether or not the agent remains capable of incorporating new information after self-edit/modification/improvement.
The paper describes that there could be failures due to representation, learning sensitivity, and meta-decision, in addition to reward hacking.
The paper proposes several directions, such as empirical metrics to measure teachability and intervention through hypothesis space and model plasticity.
There are several outlined alternative views: controllability is safer than teachability, teachability as online learning, tradeoff between performance and plasticity, usage of constitutional AI, and behavioural read-teaming.

**Position:**

Yes

**Position In Title:**

Yes

**Related Work:**

2

**Strengths And Weaknesses:**

## Strengths
- The paper provides a new perspective on measuring alignment of a changing agent during deployment.
- The paper provides a possible theoretical framework to characterize possible failure modes of teachability.

## Comments
- It is not clear why "a safe SMA is not one that is currently aligned; it is one that remains alignable." There can be cases where an agent never loses teachability but also never reaches to sufficient performance. If the message is that we want both, which I believe is the case from Section 6.3, then I believe the paper should outline this more explicitly.
  - I also fail to see evidence from the paper that the current metric fails to measure models suffering from the teachability problem.
- The three types of collapse in the taxonomy of structural incorrigibility are raised without any evidence. Even if these are possible failure modes, there is no evidence showing that they indeed happen in practice.
- It is not clear to me how this idea differentiates from continual learning. As the paper suggests using plasticity as a way to measure teachability, can the paper clarify the differences?
  - I believe the fisher-information audits is similar to techniques where the learner monitors the Hessian of the gradient [1].
  - Related, I believe even for non-self-modifying agents they can still suffer from teachability due to loss of plasticity. How does this work decouple the issues from the two?

## Typos
- Page 3, left column, line 152: Citation format should be "Wang et al. (2025a)"?
- Page 3, right column, line 137: Is ULT utility-learning tension? The acronym is never introduced.
- Missing references
  - Page 4, right column, line 216
  - Page 5, left column, line 242

## References
- [1]: Lewandowski, A., Tanaka, H., Schuurmans, D., & Machado, M. C. (2023). Directions of curvature as an explanation for loss of plasticity. arXiv preprint arXiv:2312.00246.

**Support:**

2

---

> ### Author Rebuttal · Authors · 2026-03-31
>
> We thank the reviewer for the careful reading and thoughtful engagement. Below, we respond directly to each of the main concerns and clarifications.
>
> 1. **“It is not clear why ‘a safe SMA is not one that is currently aligned; it is one that remains alignable.’”** We are happy to clarify this. Our point is that, for self-modifying agents, safety requires both acceptable current behavior and preserved future correctability. We are not claiming that teachability without competence is sufficient, or that short-horizon control is unimportant. Rather, current behavioral alignment alone is not a sufficient long-horizon invariant once the learner-state itself can change.
>
> 2. **“I fail to see evidence from the paper that the current metric fails to measure models suffering from the teachability problem.”** Our intended claim is not that existing behavioral evaluations are useless, but that they certify a different object. Behavioral evaluations are time-local properties of the current policy on a finite test distribution, whereas teachability concerns whether the post-modification learner remains correctable under bounded future intervention. For SMAs, behavior-only evaluation can validate the current policy without certifying that future corrective signals will still have leverage after self-edits. We also have a small synthetic study illustrating this separation in simplified representational and algorithmic self-modification settings, and we will reference it more clearly in the final version.
>
> 3. **“The three types of collapse in the taxonomy of structural incorrigibility are raised without any evidence.”** We respectfully disagree that the taxonomy is introduced without grounding. The Type I/II/III decomposition is not an ad hoc list of speculative failures; it is derived from the formal Utility–Learning Tension framework cited in the paper, which analyzes how self-modification over distinct components of the learner-state can erode the statistical conditions required for future correction. In this position paper, our goal is not to re-prove those underlying results, but to translate that formal insight into an alignment-facing taxonomy over the representation, update, and meta-decision axes. We agree, however, that the final draft should make this provenance clearer and distinguish more explicitly between formal grounding and empirical demonstration in deployed systems, and are happy to do so in revision.
>
> 4. **“It is not clear to me how this idea differentiates from continual learning.”** We agree this distinction should be stated more crisply. Our claim is not that self-modification defines a wholly separate formal paradigm from online or continual learning; some algorithmic self-edits can still be described within those settings. The distinction is instead about the object of safety analysis. Continual learning studies adaptation under changing experience while the overall learning mechanism is treated as fixed or externally specified. Our notion of teachability asks whether a self-modifying learner remains corrigible when it can alter the mechanisms governing future learning, especially the representational hypothesis space, update rule, and meta-decision layer.
>
> 5. **“I believe the Fisher-information audits is similar to techniques where the learner monitors the Hessian of the gradient [1].”** We appreciate this pointer and agree that Fisher-based audits are related to curvature- and Hessian-based diagnostics of plasticity loss. Our intent was not to present Fisher information as a novel or exclusive mechanism, but as one candidate proxy within a broader teachability-measurement agenda. We will incorporate this citation and clarify that the paper points to a broader class of sensitivity and hardening diagnostics.
>
> 6. **“Related, I believe even for non-self-modifying agents they can still suffer from teachability due to loss of plasticity. How does this work decouple the issues from the two?”** We agree that loss of plasticity is not unique to self-modifying systems, and the paper does not intend to deny that overlap. Our claim is narrower: in self-modifying agents, this becomes a sharper safety issue because the agent can self-edit the learner-state in ways that preserve or improve utility while reducing future corrective leverage. In that sense, self-modification makes the problem endogenous, instrumentally selectable, and potentially recursively compounding.
>
> 7. **“Typos / acronym / missing references.”** We thank the reviewer for noting these presentation issues. We will clean up the redundant citation rendering, explicitly introduce “Utility–Learning Tension (ULT)” on first use, and fix the citation LaTeX typo in the final draft.
>
> We hope these clarifications address the reviewer’s concerns, and we respectfully ask that the reviewer reconsider the score in light of the paper’s intended scope and the revisions we commit to make.

---

> > ### Author Rebuttal · Reviewer_1J2z · 2026-04-02
> >
> > Besides (3) and (4), the remaining points are addressed fully and I am happy to increase the score in general:
> > - For (3), I agree that the theory is already provided and I didn't intend to mean theoretical evidence. There are many scenarios where theoretical results fail to predict empirical observations for various reasons: strict assumptions, vacuous bounds, etc. It would be nice to have empirical evidence but even without this, I now lean toward accept.
> > - For (4), I'm not 100% sure if the distinction is clear to me. For example, do [1] and [2] fall into the former where the "learning mechanism" is fixed? Wouldn't the proposed mechanism in the paper also be a fixed mechanism?
> >
> > References:
> > - [1]: Hernandez-Garcia, J. F., Dohare, S., Luo, J., & Sutton, R. S. (2025). Reinitializing weights vs units for maintaining plasticity in neural networks. arXiv preprint arXiv:2508.00212.
> > - [2]: Dohare, S., Hernandez-Garcia, F., Lan, Q., Rahman, P., & Mahmood, R. Dynamic Deep Learning.

---

### Decision · Program_Chairs · 2026-04-30

**Decision:**

Accept (regular)

**Comment:**

This submission addresses an important problem of how we should approach safety in self-modifying agents, which is a timely topic. Reviewers generally agree that the paper makes a valuable conceptual contribution and starting point for future research in this direction. There were some concerns that there is limited empirical grounding of the position and some distinctions need clarity (e.g. continual learning), but these were adequately addressed in the rebuttal period. Overall, I vote accept.